| Editor's Pick | Conduct of Scientific Research | Perspective

# Building a queer- and trans-inclusive microbiology conference

Rachel Gregor,[1] Juliet Johnston,[2] Lisa Shu Yang Coe,[3,4] Natalya Evans,[5] Desiree Forsythe,[6] Robert Jones,[7] Daniel Muratore,[8] Bruno Francesco Rodrigues de Oliveira,[9] Rachel Szabo,[1] Yu Wan,[10] Jelani Williams,[11] Callie R. Chappell,[12] Shayle B. Matsuda,[13] Melanie Ortiz Alvarez de la Campa,[14] Queer and Trans in Microbiology Consortium, J. L. Weissman[6,11]

**ABSTRACT** Microbiology conferences can be powerful places to build collaborations and exchange ideas, but for queer and transgender (trans) scientists, they can also become sources of alienation and isolation. Many conference organizers would like to create welcoming and inclusive events but feel ill-equipped to make this vision a reality, and a historical lack of representation of queer and trans folks in microbiology means we rarely occupy these key leadership roles ourselves. Looking more broadly, queer and trans scientists are systematically marginalized across scientific fields, leading to disparities in career outcomes, professional networks, and opportunities, as well as the loss of unique scientific perspectives at all levels. For queer and trans folks with multiple, intersecting, marginalized identities, these barriers often become even more severe. Here, we draw from our experiences as early-career microbiologists to provide concrete, practical advice to help conference organizers across research communities design inclusive, safe, and welcoming conferences, where queer and trans scientists can flourish.

**KEYWORDS** equity, inclusion, LGBTQ, queer and trans, scientific conferences

Microbiology conferences serve as opportunities for microbiologists to engage with colleagues, learn new research, and build community. For queer and transgender microbiologists, the excitement of attending a conference is often tempered by previous experiences of exclusion and discrimination in professional spaces . Over the past year, initiatives have been developed by queer and trans community members to improve conditions at conferences, including at the 2022 Marine Microbes Gordon Research Conference (GRC) and International Society of Microbial Ecology (ISME) 2022 meeting, as well as through in-person and virtual meetings organized by the Society Champions group in the Microbiology Society. Here, we, a team of queer and trans scholars and microbiologists involved in these initiatives (see S1 Text for a positionality statement), contribute our experiences navigating professional conferences to provide guidance on making these spaces more inclusive (Fig. 1). While many conference organizers have been enthusiastic about these grassroots efforts, and some of the suggestions below have already been implemented, we aim to provide a more comprehensive guide which may serve as guidance across disciplines. Our goal is to continue these conversations, so they permeate throughout all levels of leadership and ensure the success of the next generation of microbiologists.

## Why focus on queer and trans scientists?

The attrition of queer and trans trainees from the sciences is well documented—lesbian, gay, bisexual, queer, trans, and/or gender nonconforming (LGBTQ+) undergraduates are 8%–10% less likely to persist in science, technology, engineering, and math (STEM) than heterosexual and/or cisgender students (1, 2) (see Box 1 for a note on language). Many studies have revealed that STEM environments are often heteronormative, anti-feminine,

Address correspondence to Rachel Gregor, rgregor@mit.edu, or J. L. Weissman, jw4336@terpmail.umd.edu.

Rachel Gregor, Julie Johnston, and J. L. Weissman contributed equally to this article. Author order was determined both by additional roles as corresponding authors and by increasing seniority (due to differing preferences for first/last position based on career stage).

Lisa Shu Yang Coe, Natalya Evans, Desiree Forsythe, Robert Jones, Daniel Muratore, Bruno Francesco Rodrigues de Oliveira, Rachel Szabo, Yu Wan, and Jelani Williams contributed equally to this article. Author order was determined alphabetically.

Callie R. Chappell, Shayle B. Matsuda, and Melanie Ortiz Alvarez de la Campa contributed equally to this article. Author order was determined alphabetically.

The authors declare no conflict of interest.

See the funding table on p. 10.

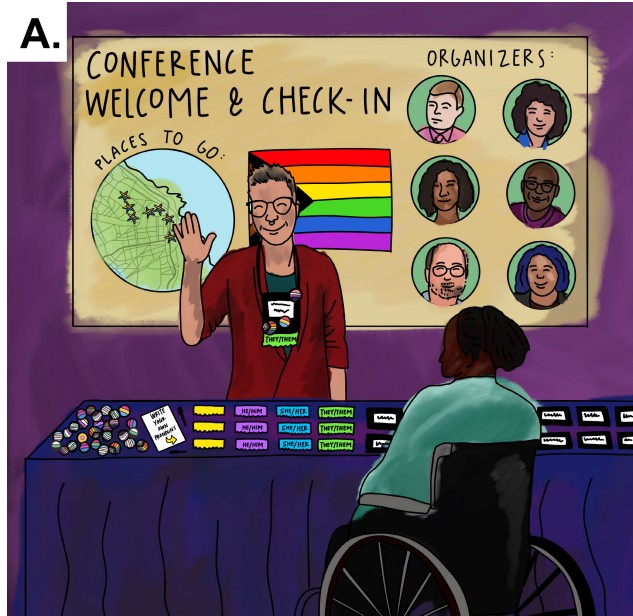
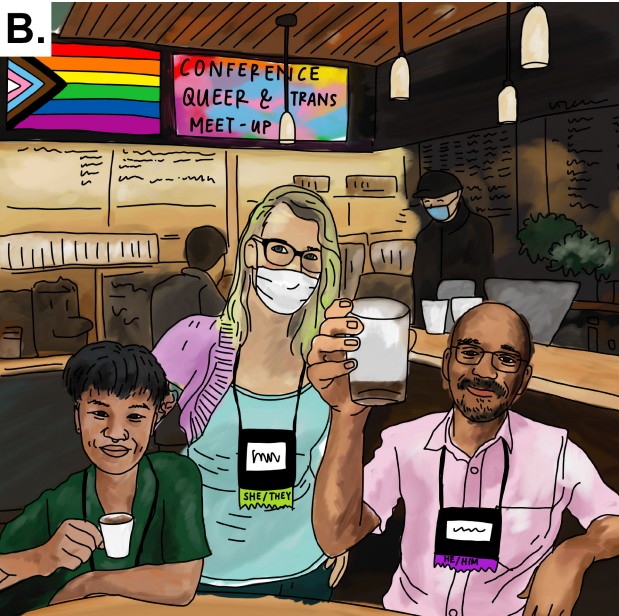
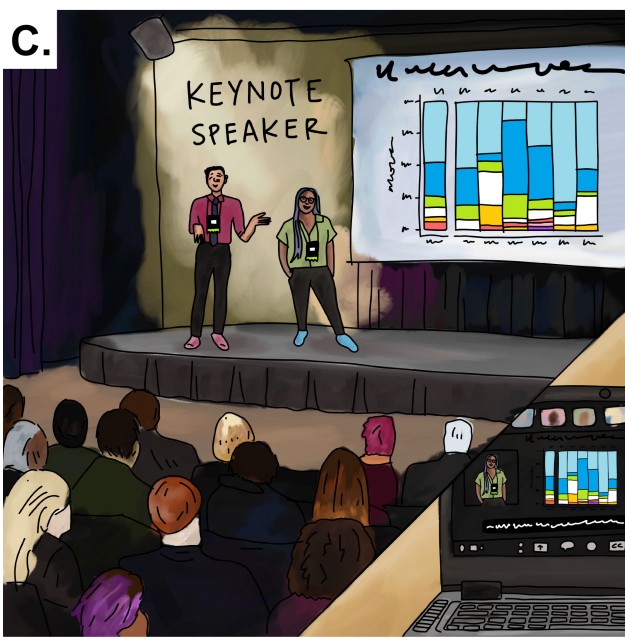
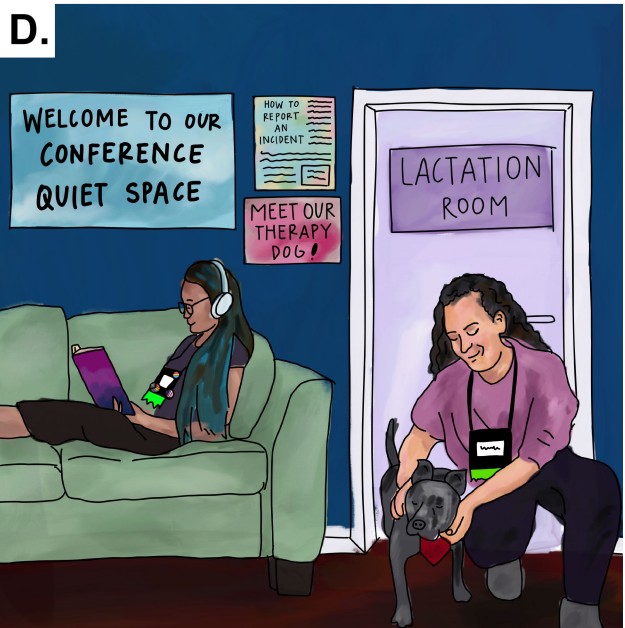

**FIG 1** What does an inclusive conference look like? Some recommendations include: (A) pronoun ribbons, including fill-in-the-blank options (see "Registration and demographic data"), (B) queer and trans networking events (see "Networking and mentorship programs"), (C) hybrid attendance options (see "Virtual and hybrid options"), and (D) designated spaces to support attendees and meet their needs (see "Physical spaces"). Illustration by Callie Rodgers Chappell.

and toxically competitive (3–5), and LGBTQ+ students face significant marginalization and increased depression, exhaustion, and general stress relative to their peers (6). For queer and trans scientists with other intersecting identities, for example, relating to disability, first-generation student status, socioeconomic status, and/or marginalized race or ethnicity, these various axes of marginalization can compound (7), leading to feelings of invisibility (8, 9). While our discussion below centers on queer and trans scientists, our recommendations have significant benefits across multiple identities such as quiet spaces for neurodiverse attendees, and local climate resources related to race and ethnicity. Increasing inclusivity efforts benefits everyone.

---

**BOX 1. A NOTE ON LANGUAGE**

We use queer as a term meant to include all people of marginalized sexualities and genders who are not heterosexual and/or cisgender (i.e., a person's gender aligns with the sex assigned to them at birth). We recognize that acceptable labels vary across individuals and over time and that no one word or acronym best captures the diverse identities and experiences within our community. We highlight trans and nonbinary identities in particular as a further marginalized community within an already marginalized community.

While we focus on queer and trans folks, we understand this (our) community encompasses a large array of intersecting identities that face additional and unique challenges. Community members who are part of marginalized races and ethnicities or are disabled, for example, may experience different and additional barriers in conferences that should be further considered by the organizing committee. These topics cannot be fully given the length of discussion they deserve within the allotted space for this perspective.

---

Scientific conferences provide opportunities for scientific and professional education, as well as networking that is critical to career growth. However, discrimination based on gender and sexual orientation continues to permeate professional spaces in the sciences, preventing queer and trans scientists from fully engaging in conferences without compartmentalizing or censoring their identities (10–13). Furthermore, a "just-focus-on-the-science" culture prevalent in these spaces dissuades marginalized groups from vocalizing the issues they face, thus stifling progress toward inclusivity and diversity (14). This status quo need not persist and is damaging to the field. Below, we outline practical suggestions for organizing committees to implement before, during, and after conferences, to create an environment that is safe and welcoming for queer and trans scientists (Table 1).

## BEFORE THE CONFERENCE

### Venue and location

Costs, legal protections, and cultural norms are important considerations in selecting an inclusive conference venue. High-cost venues disproportionately discourage queer and trans attendees, who are more likely to face financial precarity (15, 16), meaning they may not be able to afford upfront payments for later reimbursement as is common in many academic institutions. Countries continue to pass egregious new anti-LGBTQ+ laws (17, 18), including the USA, where anti-trans legislation is being passed at a record rate (19, 20; see ACLU map of legislation: https://www.aclu.org/legislative-attacks-on-lgbtq-rights). This challenges the status quo that prioritizes the USA, Canada, UK, and EU as "queer-friendly" locations and ignores significant subregional variations in legal, racial, and socioeconomic challenges. Ultimately, this view neglects ongoing safety concerns, excludes participants from low- and lower-middle-income countries who already face disproportionate financial barriers due to visa requirements, and maintains a hierarchy between colonizers and colonized.

Consult with local queer and trans organizations during planning for nuanced and informed views regarding legislation and safety risks queer and trans attendees may face and present this information to attendees before the conference (see example in S3 Text). Offer travel awards for queer and trans attendees with additional priority to international and/or lower socioeconomic status attendees. Awards based solely on financial need are likely to help queer and trans attendees as well, but may not be sufficient to counteract the various barriers these attendees face internationally and in academia that may not always be immediately visible. Ensure venues have appropriate accessibility features (e.g., ramps, quiet rooms, elevators, etc.)—this intersectional

**TABLE 1**  LGBTQ+ inclusivity checklist[a]

| Item | Description |
|------|-------------|
| Organizing committee | • Stay up to date on fundamental inclusivity practices (see S2 Text)<br>• Recruit a diverse organizing committee from the outset of planning<br>• Ensure that there is a designated contact for any accessibility and inclusion questions or concerns |
| Venue selection | • Consider financial barriers for attendees when choosing a venue<br>• Consider the history of specific businesses in being welcoming<br>• Consider if all-gender restrooms are already available at the venue or if the venue will allow their designation during the conference<br>• Consider the venue's accessibility features for people with disabilities<br>• Consider the venue's COVID-19 policies and willingness to implement COVID safety precautions |
| Local climate | • Create materials explaining any potential safety concerns for attendees around, for example, transphobia, racism, ableism, and misogyny. Even locations that are popularly viewed as queer and trans friendly often have the potential to quickly become unsafe<br>• Seek out guidance from local scientists from representative groups around these issues, including whether it is advisable to hold a conference in a given location |
| Registration design | • Do not require name submission to be a legal name; if a legal name must be collected, provide an additional field for the name, since these may not overlap for all attendees<br>• Provide fill-in-the-blank pronoun options<br>• Ensure names and pronouns are printed correctly on all conference materials<br>• If collecting gender or demographic information, ensure privacy and transparency about how data will be used |
| Information disclaimer | • Create a brief guide explaining when registration information will be used for housing or attendance statistics |
| Code of conduct | • Design a code of conduct with community input and highlight that code during opening ceremonies<br>• Make sure the code of conduct has specific steps for enforcement with appropriate resources to enact this enforcement<br>• Be explicit in the kind of conduct that will not be tolerated at this conference |
| Confidential reporting | • Advertise how to confidentially report potential harassment and potential follow-up actions<br>• Hire staff with expertise and training in handling confidential and sensitive reports<br>• Ensure there is a direct and actionable plan for participants' safety in cases of harassment<br>• Have explicit policies and mechanisms protecting those reporting offenses from retaliation |
| Name tags | • Provide pronoun pins or badge ribbons, including "fill-in-the-blank" style ones |
| Shared housing | • Provide some single rooms, optional self-identification for queer and trans participants, and exchange contact information between roommates ahead of the conference<br>• Suggest and provide links to find alternate, affordable accommodations if single rooms are not available at the official conference lodging<br>• Let participants pick their roommates when possible and do not restrict different-gender pairings<br>• Plan for any issues of discrimination or harassment by designating a point of contact and reserving spare rooms |
| Gender-neutral language | • Use gender-neutral terminology and avoid strictly binary language (see Table 2)<br>• Proofread all forms and practice introductions, including name pronunciations<br>• Avoid "ladies and gentlemen"; see Table 2 for alternatives |
| Gender-neutral and gendered bathrooms | • Advertise the location of all-gender bathrooms<br>• Ensure all single-stall restrooms are marked as gender neutral<br>• Advertise that all attendees are welcome to use the restrooms that align with their gender and that no individual should ever be harassed or required to provide documentation of their gender identity to access restrooms |
| Quiet space | • Reserve a dedicated room at the conference venue to relax in silence<br>• Dim the lights and minimize sounds. Bonus to add pillows or blankets for comfort and to provide water and snacks |
| Mentorship | • Create an identity-based mentorship pairing option where there is the option to match queer and trans mentor/mentee pairs |
| Conference workshops and networking events | • Create programming and designate funding for specific groups of marginalized attendees to gather, form networks of support, and plan together<br>• Provide funding for existing affinity groups and organizations<br>• Provide networking spaces that enforce COVID safety precautions (e.g., masking, social distancing) |
| Follow-up statistics | • Compile transparent reports on conference attendance and inclusivity efforts and impacts<br>• Reference past years to demonstrate progress and improvement goals<br>• Make all reports and aggregated statistics publicly available |
| Use an intersectional approach | • Do not take the experiences of white queer and trans participants as the default or universal experience of all queer and trans participants<br>• Recognize that attendees with multiple, overlapping marginalized identities may face distinct challenges in your field/at your venue<br>• Acknowledge that within the queer community, there is a great diversity of experiences that cannot and should not be reduced to a single perspective |

[a]See relevant sections of text for further discussion of each point.

issue affects many queer and trans people, who experience an increased proportion of disabilities and chronic illnesses (21 [note that this study frames disability as a "public health epidemic"—a problematic stance on disability based on the medical model—nevertheless, information regarding the prevalence of disability in the LGBTQ+ community is important due to its scarcity of reporting in other primary literature]).

When explicit laws and policies put queer and trans attendees in danger, seek alternative venues. However, do not uncritically avoid entire regions based on "queer friendliness" rankings, as this abstracts away from important local variability and reifies racist biases.

## Steering committee preparation

The steering committee should be a diverse group of individuals with a range of perspectives, including queer and trans members. In particular, create space for queer and trans organizers of color since white voices represent a small and biased fraction of the community. Conference organizers, session leaders, and support staff should explicitly discuss the issues outlined here as well as basic trainings on queer and trans inclusion (S2 Text), sensitivity, and gender-inclusive language (Table 2) (22, 23). Ensure that there is a point person clearly indicated on the conference website to be contacted for any accessibility and inclusion needs and concerns.

## Registration and demographic data

### Names

There should always be an option to provide a name separately from "legal names" due to barriers for trans and queer people to update their names. Always include a "name" option accompanied by text indicating that it will be used in conference materials, printed name badges, and registration. If a legal name is required, for example for visa purposes, or if a professional name or ORCID is required for published conference proceedings, include these as separate options accompanied by explanations.

### Pronouns

An optional space (never required) to provide fill-in-the-blank pronouns during registration offers flexibility for trans and nonbinary attendees to choose how to present themselves and encourages cis attendees to acknowledge their own gender (see S2 Text for guides on how to learn and use correct pronouns for your peers). Pronouns should be carried over to conference materials. Pronoun stickers/badges with fill-in-the-blank options can be made available for attendees to write in pronouns. Pronouns are independent of gender and should not be assumed.

### Demographic data

While collecting data on gender identity and sexuality can provide insight and visibility for the LGBTQ+ community (24), outing queer and trans scientists can put them at significant personal and professional risk. Therefore, consider which demographic data are necessary and explicitly state the intended purpose of the data and how they are disseminated, and provide safeguards for anonymity and data storage.

Fill-in-the-blank fields are best for reporting gender and sexuality and avoiding "othering" terminology (25, 26); if categories must be pre-determined, provide the option to check multiple boxes. Remember that "transgender" should not be listed as a gender in and of itself (i.e., trans women are women and trans men are men).

## Accommodations

Registration paperwork should outline procedures for housing (if available) which includes single rooms and self-organized pairs/groups, regardless of gender, to accommodate trans and nonbinary attendees (27–29). Reserve spare rooms and provide

**TABLE 2** Common examples of gender-exclusive language that can be replaced with gender-inclusive alternatives[a]

| Commonly used exclusive language | Inclusive alternatives | Additional notes |
|---|---|---|
| • Ladies and gentlemen<br>• Men and women | • Everyone<br>• Colleagues<br>• Participants<br>• Attendees<br>• Audience members<br>• People<br>• Folks/folx<br>• Team<br>• You all | Avoid making and voicing assumptions about any individual's gender or the gender composition of an audience when addressing or referring to people.<br>Cis and trans women are women, and cis and trans men are men. It is invalidating and unnecessary to distinguish between cis and trans people when referring to their genders unless (i) the conversation intentionally involves discussing or acknowledging trans and gender nonconforming communities or (ii) an individual self-identifying as trans or gender nonconforming chooses to discuss their identity. |
| • Either gender<br>• Both genders<br>• The opposite gender<br>• The other gender | • Any or no gender<br>• All genders<br>• Another gender<br>• A different gender | Agender people do not identify as women, men, or some combination thereof, but instead do not have a gender at all. Genderfluid and nonbinary people do not cleanly fall into this binary and may identify as multiple genders. |
| • Wife/husband; girlfriend/boy-friend<br>• Maternity/paternity leave<br>• Pregnant women | • Partner; spouse<br>• Parental leave<br>• Pregnant people | Referring only to "women" in the context of parental leave and pregnancy excludes trans men and nonbinary/gender-nonconforming people. These suggestions are for when speaking in general terms. Individuals may, of course, refer to themselves and their relationships using the terms that are most appropriate for them. |
| • Ladies'/men's room | • Restroom<br>• Bathroom<br>• WC<br>• Toilet | |
| • Preferred/chosen pronouns/gen-der/name<br>• "What are your preferred pronouns?" | • Pronouns, gender, name<br>• "Which pronouns do you use?" | Describing someone's pronouns, gender, or name as "preferred" or "chosen" is invalidating, implying that respecting someone's gender identity is optional rather than necessary.<br>If someone's pronouns are unknown, it is okay to use "they/them/theirs" as gender-neutral options before learning which pronouns they use, rather than defaulting to gendered pronouns based on someone's appearance. |
| • "As women/men, we…" | • "In my experience, as a ___, I…" | When sharing a personal experience with the intent of connecting with an audience, a speaker can avoid voicing incorrect assumptions about the identities and experiences of the listeners using an alternative phrasing like this one. |

[a]We acknowledge that these words and phrases are commonly used in countries in North America and Europe and predominantly English-speaking communities; evolving gender-inclusive language practices may vary by region and culture. In addition, linked resources for additional languages can be found here: https://nonbinary.wiki/wiki/Gender_neutral_language.

a contact for queer and trans attendees who may face discrimination or harassment based on cohabitation. In addition, offer to assist disabled attendees to find accessible lodging.

## Virtual and hybrid options

Virtual and hybrid conferences reduce socioeconomic and environmental costs and increase accessibility for many marginalized groups, including queer and trans scientists (30). Similar to in-person conferences, it is critical to follow best practice guidelines for creating and maintaining welcoming virtual spaces (31), including a clear code of conduct (32), timely responses to unacceptable behaviors, and participants' consent to posting materials on the conference's official social media (33). Conference organizers should also maintain a reactive support team to ensure participants' profiles are correctly displayed and address other technical issues, and the conference safety team should be trained to recognize and respond to virtual forms of harassment.

## DURING THE CONFERENCE

### Atmosphere and language

Create a welcoming space and set the tone during opening ceremonies by establishing the code of conduct and mentioning accommodations for queer and trans participants. Organizers and session leaders should consciously use gender-inclusive terminology (Table 2) and avoid assuming attendees' genders. These meaningful actions establish gender-inclusive norms for everyone. Conference materials should include resource guides so that community members are up to date on inclusive practices (S2 Text). Be mindful when speaking broadly about gender diversity, especially when topics may or may not be limited to particular groups (e.g., "pregnant scientists" is more inclusive than "women"). It is important to consider that not all attendees will be familiar with this language, particularly those who are not native English speakers, and to provide support and patience in the process of adopting these terms.

### Physical spaces

#### All-gender restrooms

All-gender restrooms should be available and accessible to persons with disabilities. Clearly mark these restrooms in conference maps, apps, and during opening session announcements. Similarly, nongendered lactation spaces should be made available to those currently breastfeeding. Reinforce that all attendees are welcome to use the restrooms that align with their gender and that no individual should ever be harassed or require documentation of their gender identity to access restrooms.

#### Quiet spaces

On-site dedicated quiet spaces are vital to help attendees cope with socially stressful situations and maintain professionalism without leaving the venue. Everyone can benefit from quiet spaces, including marginalized groups such as queer and trans scientists who often face microaggressions in professional settings (34) or neurodiverse attendees who are overstimulated. Encourage everyone to use quiet spaces to recenter and recharge, while establishing it is not a workspace.

#### Family policies

Family-inclusive policies (e.g., attendance passes for spouses, childcare) must include same-sex partners and nonnuclear family structures. Due to the lack of LGBTQ+ legal recognition in many places, documentation for accommodations should not be required.

### Networking and mentorship programs

Develop a network of scientists with shared backgrounds for early career researchers to find support and advice and help create community in marginalized groups, which is linked to stronger scientific identity and retention in STEM (35–37). Networking can be facilitated by planning queer and trans social and professional events, including for specific groups such as queer and trans People of Color (35–37). For example, in 2021–2023, LGBTQ+ events have been held at Marine Microbes GRC, ISME, American Society for Microbiology Microbe, the Australian Society for Microbiology Annual National Conference, the Australian Microbial Ecology annual meeting, the Society for Integrative and Comparative Biology, the Microbiology Society Annual Conference, and the World Microbe Forum, to name a few. In addition, the Pride in Microbiology Network has been recently founded to provide a platform for these events year-round (37).

Wide advertisement of LGBTQ+ events is critical, as it may not always be apparent which conference attendees are queer and/or trans, and attendees may struggle to connect with their community otherwise. It is important to note explicitly which events are for specific groups in particular and which events allies would be encouraged to

attend. Privacy can be a concern for attendees who participate in these events; therefore, ask for photo consent and clearly state whether photos will be taken and publicized. Consider off-site events at inclusive spaces for attendees' safety and privacy, in consultation with local queer and trans organizers to ensure that any local community spaces are respected.

Mentorship programs help trainees and first-time conference attendees take full advantage of their experience (38). We recommend optional identity-based mentorship, which can be arranged before a conference, to promote increased trust and authentic engagement between mentors and mentees (39). A notable example is Binning Singletons (38) at the American Society for Microbiology's Microbe conference, which fostered a high percentage of LGBTQ+ participants leading to dedicated LGBTQ+ meetups at future events.

## Safety

Queer and trans participants are particularly vulnerable to identity-based and sexual harassment (28, 40–42); therefore, simple and clear reporting systems must be made accessible. These mechanisms and policies should be described in the conference materials, opening session, and information desk, and include anonymous reporting (e.g., through a conference app or web form). The outcomes of both confidential and nonconfidential reporting mechanisms should be made clear, as well as mandated reporting requirements.

Reporting mechanisms are especially important for events with alcohol, which increases the likelihood of inappropriate behavior and harassment. Consider hosting alcohol-free events for safety and to benefit sober participants, and always provide multiple nonalcoholic options at all events.

Designate a crisis response team with sensitivity to marginalized communities' experiences that includes queer and trans individuals with varied identities. Consider hiring a third-party mediator to handle code of conduct violations. Do not equate police presence with safety, as queer and trans communities are often the subjects of police harassment (43). In addition, monitor social media and conference hashtags for harassment of specific conference attendees or marginalized groups.

## COVID-19 safety precautions

Minimize the risk of COVID-19 and other transmissible illnesses and provide an inclusive space for disabled and immunocompromised attendees with precautions like hybrid attendance (44), masking, venues with outdoor spaces, and dedicated networking spaces with air purifiers (additional resources in S2 Text).

## AFTER THE CONFERENCE

### Survey data

Demographic data collection should follow the best practices outlined above. Free-form responses are particularly valuable, especially explicit questions about queer and trans experiences, to gain insight into the effectiveness of any inclusion initiatives. Share an aggregate narrative report of survey results, while preserving participants' anonymity. Keep in mind that small sample sizes may make anonymity impossible; this should be noted and the data presented with caution.

### Transparent reporting

Generate a comprehensive post-conference report to evaluate progress, identify challenges, hold organizers accountable, and track ongoing efforts. Public reports which highlight and describe attendance and inclusive programming (e.g., mentorship programs, events) will encourage prospective queer and trans participants to attend future conferences.

Permanent structures within organizations are especially effective for implementing change and gaining feedback. For example, the Microbiology Society has a Members Panel that voices issues from marginalized groups, including the LGBTQ+ community (see https://microbiologysociety.org/why-microbiology-matters/council-governance/standing-panels/members-panel.html). These sub-groups advise and participate in events' organizing committees, oversee past implementation, and track progress, and should be given structural power and/or compensation when possible.

## CONCLUSIONS AND OUTLOOK

We emphasize necessary policies and practices which support the increasing proportion of early-career researchers who present themselves authentically in professional communities. Many efforts are already underway by organizing committees to provide better conference experiences for marginalized groups, and we urge others to strive for the same. We recognize that it might not always be possible to implement all of these recommendations or that some may occasionally be at odds with each other, requiring organizers to implement creative solutions in collaboration with queer and trans stakeholders from their research community. Nevertheless, we believe that the scope of these best practices is within reason for many conferences, and we strongly encourage organizers to prioritize these issues and continue in dialogue with queer and trans community members as needs and norms evolve in the future.

To conclude, we highlight two pieces of advice that conference organizers should keep in mind: First, you need to have a diverse organizing committee representing the groups you wish to include. These community members should be included as valued members from the very beginning of the planning process, not as a last-minute addition. Second, recognize that people with multiple, overlapping marginalized identities may face unique and magnified challenges that other attendees may not, especially when intersecting identities vary by cultural context (e.g., inequality of queer and trans People of Color can be magnified country-to-country) (7). Inclusivity strategies need to be designed around the experiences of the most marginalized, not limited to white queer and trans participants, to broaden inclusion at conferences.

## ACKNOWLEDGMENTS

All authors are members of the Queer and Trans in Microbiology Consortium. The Queer and Trans in Microbiology Consortium also includes the following persons: Anna C. Fagre (Colorado State University); Steven A. Frese (University of Nevada, Reno); Maria Hamilton (University of Georgia); Maurizio Labbate (Australian Society for Microbiology); McK Mollner (University of Maine); Itumeleng Moroenyane (Stellenbosch University); Leonardo Pacciani-Mori (University of California, San Diego); Gonçalo J. Piedade (NIOZ Royal Netherlands Institute for Sea Research and University of Amsterdam); Sammy Pontrelli (ETH Zurich); Rahmeen Rahman (Imperial College London); Heema Kumari Nilesh Vyas (The University of Sydney); Mica Y. Yang (Stanford University); Anna C. B. Weiss (University of Southern California); and Olivier Zablocki (The Ohio State University).

All authors thank the GRC, ISME, and Microbiology Society leadership for their support and for providing a platform for these conversations to take place. We thank the following colleagues for their feedback on the manuscript: Jana S. Huisman (Massachusetts Institute of Technology), Vilhelmiina Haavisto (ETH Zurich), Elizabeth N. Rudzki (University of Pittsburgh), and Pedro Humberto Lebre (University of Pretoria). R.G. is grateful to Lauryn McNair (Massachusetts Institute of Technology) for initial guidance and input.

J.J. is grateful for the support of the American Society for Engineering Education eFellows Postdoctoral Research Fellowship funded through the National Science Foundation (grant #EEC-2127509). R.G. and J.L.W. are grateful for support from the Simons Foundation Marine Microbial Ecology Postdoctoral Fellowships (R.G.: 653410; J.L.W.: 653212). R.G. is grateful for the support of the Center for Chemical Currencies of

a Microbial Planet Postdoctoral Fellowship (National Science Foundation, OCE-2019589; this is the NSF Center for Chemical Currencies of a Microbial Planet (C-CoMP) publication #023). D.M. is funded by the S. F. I. Omidyar Postdoctoral Fellowship. Y.W. is an Imperial Institutional Strategic Support Fund Springboard Research Fellow, funded by the Wellcome Trust and Imperial College London. Y.W. thanks the support from the National Institute for Health Research Health Protection Research Unit in Healthcare Associated Infections and Antimicrobial Resistance at Imperial College London in partnership with the UK Health Security Agency, in collaboration with Imperial Healthcare Partners, University of Cambridge, and University of Warwick. B.F.R.D.O. is grateful for the support of an Office of Naval Research (ONR) grant (award #: N62909-23-1-2021) and Fundação de Amparo à Pesquisa do Estado do Rio de Janeiro (FAPERJ) grant (award #: E-26/211.284/2021). The views expressed in this publication are those of the author(s) and not necessarily those of the UK N.H.S., the National Institute for Health Research, the Department of Health and Social Care or the UK Health Security Agency, or the U.S. federal government and its agencies.

## AUTHOR AFFILIATIONS

[1]Department of Civil and Environmental Engineering, Massachusetts Institute of Technology, Cambridge, Massachusetts, USA

[2]Georgia Institute of Technology, School of Civil and Environmental Engineering, Atlanta, Georgia, USA

[3]Division of Science, Biology Program, New York University Abu Dhabi, Abu Dhabi, UAE

[4]Department of Biology, University of Florida, Gainesville, Florida, USA

[5]Marine Science Institute, University of California, Santa Barbara, California, USA

[6]Schmid College of Science and Technology, Chapman University, Orange, California, USA

[7]U.S. Army Engineer Research and Development Center, Cold Regions Research and Engineering Laboratory, Hanover, New Hampshire, USA

[8]Santa Fe Institute, Santa Fe, New Mexico, USA

[9]Department of Microbiology and Parasitology, Biomedical Institute, Fluminense Federal University, Niterói, Rio de Janeiro, Brazil

[10]NIHR Health Protection Research Unit in Healthcare Associated Infections and Antimicrobial Resistance, Department of Infectious Disease, Imperial College London, London, United Kingdom

[11]Department of Biological Sciences, University of Southern California, Los Angeles, California, USA

[12]Department of Biology, Stanford University, Stanford, California, USA

[13]Daniel P. Haerther Center for Conservation and Research, John G. Shedd Aquarium, Chicago, Illinois, USA

[14]Department of Molecular Microbiology & Immunology, Brown University, Providence, Rhode Island, USA

## AUTHOR ORCIDs

Rachel Gregor http://orcid.org/0000-0003-4071-9573
J. L. Weissman http://orcid.org/0000-0002-4237-4807

## FUNDING

| Funder | Grant(s) | Author(s) |
| --- | --- | --- |
| Simons Foundation (SF) | 653410 | Rachel Gregor |
| Simons Foundation (SF) | 653212 | J. L. Weissman |
| National Science Foundation (NSF) | OCE-2019589 | Rachel Gregor |
| National Science Foundation (NSF) | EEC-2127509 | Julie Johnston |

| Funder | Grant(s) | Author(s) |
| --- | --- | --- |
| Office of Naval Research (ONR) | N62909-23-1-2021 | Bruno Francesco Rodrigues de Oliveira |
| Fundação de Amparo à Pesquisa do Estado do Rio de Janeiro (FAPERJ) | E-26/211.284/2021 | Bruno Francesco Rodrigues de Oliveira |

## ADDITIONAL FILES

The following material is available online.

### Supplemental Material

**Supporting Information (mSystems00433-23-s0001.docx).** Author positionality statement and additional resources.

### Open Peer Review

**PEER REVIEW HISTORY (review-history.pdf).** An accounting of the reviewer comments and feedback.

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
