## [Reviewer comments · mSystems]

Building a Queer- and Trans-Inclusive Microbiology Conference

Rachel Gregor, Julie Johnston, Lisa Coe, Natalya Evans, Desiree Forsythe, Robert Jones, Daniel Muratore, Bruno de Oliveira, Rachel Szabo, Yu Wan, Jelani Williams, Callie Chappell, Shayle Matsuda, Melanie Ortiz Alvarez De La Campa, Queer and Trans in Microbiology Consortium, and JL Weissman

Corresponding Author(s): Rachel Gregor, Massachusetts Institute of Technology

Review Timeline:

Submission Date:	May 1, 2023
Editorial Decision:	June 3, 2023
Revision Received:	July 18, 2023
Accepted:	August 8, 2023

Editor: Suzanne Ishaq

Reviewer(s): Disclosure of reviewer identity is with reference to reviewer comments included in decision letter(s). The following individuals involved in review of your submission have agreed to reveal their identity: Jennifer Middleton (Reviewer #1); María Rebolleda Gómez (Reviewer #2)

Transaction Report:

DOI: <https://doi.org/10.1128/msystems.00433-23>

June 3, 2023

Dr. Rachel Gregor
Massachusetts Institute of Technology
Civil and Environmental Engineering
Cambridge, MA 02141

Re: mSystems00433-23 (Building a Queer- and Trans-Inclusive Microbiology Conference)

Dear Dr. Rachel Gregor:

Thank you for submitting your manuscript to mSystems. We have completed our review and I am pleased to inform you that, in principle, we expect to accept it for publication in mSystems. However, acceptance will not be final until you have adequately addressed the reviewer comments.

Preparing Revision Guidelines

Please return the manuscript within 60 days; if you cannot complete the modification within this time period, please contact me. If you do not wish to modify the manuscript and prefer to submit it to another journal, please notify me of your decision immediately so that the manuscript may be formally withdrawn from consideration by mSystems.

Sincerely,

Suzanne Ishaq

Editor, mSystems

Journals Department
Reviewer comments:

Reviewer #1 (Comments for the Author):

Review of manuscript:
Building a Queer and Trans-Inclusive Microbiology Conference
By Rachel Gregor et al.

Review by Jennifer Middleton

This work presents a thoughtful, well-rationed, and easy to follow set of suggestions for conference organizers to consider when aiming to host conferences that are safe and inclusive of queer and trans attendees. The suggested guidelines represent a synthesis of the authors' diverse conference experiences and build upon the growing body of literature investigating the LGBTQ+ experience more broadly. Importantly, this work gives careful consideration to intersectionality within the LGBTQ+ community and several of the recommended actions (including consideration of local climate, confidential reporting, travel awards for low-income attendees, and on-site quiet spaces) will benefit a wide-range of conference attendees outside of the LGBTQ+ community as well. Although this work is targeted for the microbiology community, many of the recommendations are easily transferrable to other disciplines. Further, this work includes in its appendix an additional compilation of helpful resources and guides that will be quite useful to anyone interested in building a more inclusive STEM environment. As such, I believe this work provides a valuable framework for queer and trans-inclusion both in the microbiology community and across STEM disciplines.

General Comments:

I appreciate the authors' efforts to motivate and produce a clear and actionable set of recommendations for queer and trans-inclusion at STEM conferences. I especially appreciate the authors' call to consider regional and cultural nuances in alleged queer-friendly spaces, patience in the adoption and use of evolving language and terminology, and attention to avoiding white-bias when considering LGBTQ+ experience. This is already a valuable and important contribution. Below I have listed a few specific suggestions to help clarify a few of the main points from the text for a wider audience.

Specific Comments:

Abstract: I appreciate that this article is written primarily by and targeted for the microbiology community. However, given the number of suggestions that are clearly applicable across STEM fields (and beyond), I recommend the authors highlight within the abstract the utility of this work for applications outside of microbiology as well.

Lines 108-109: Perhaps this point is clear to those within the microbiology community, but from an outside perspective I think it would be helpful for the authors to clarify/elaborate on how personal financial precarity affects one's ability to attend professional conferences (which in my field are typically covered by research grants or other university funds). Alternatively, if this line is meant to address professional financial security (e.g., perhaps less funding allocated to queer and trans people), then I encourage the authors to clarify this point.

Lines 119-120: If the argument here is that travel awards are needed to help queer and trans attendees to attend conferences because they are financially disadvantaged, then it is not clear to me from this article why this suggestion is for specific awards for queer and trans attendees and not a blanket suggestion for more awards for attendees facing financial difficulties. If the argument is that specific awards for queer and trans people are necessary to increase queer and trans attendance (in the case that blanket financial awards typically overlook the queer and trans community, for example), then I believe that point needs to be better clarified.

Lines 124-126. This is an important point and I appreciate the authors noting the nuance here.

Lines 216-220: It might be worth making the point here that it can be difficult (or may feel unsafe) for queer and trans scientists to organically meet and identify each other in the professional environment outside the context of a dedicated event because this aspect of identify is not necessarily visually apparent and may not come up naturally in conversation.

Reviewer #2 (Comments for the Author):

This is a great compendium of best practices to increase inclusivity for queer and trans people in scientific conferences. At a time of intense backlash against queer and trans communities, it is a pressing piece. The paper is clear, holistic, and takes intersectionality seriously. As such, it is an important contribution that can be very helpful for committees organizing conferences.

My only comment in terms of improving the manuscript is to potentially acknowledge that these are best practices and that in some cases some might be in conflict. Having organized conferences outside of Europe and North America, I can say that sometimes the spaces available might not support a hybrid set-up and economic constraints might restrict the available extra spaces (that is, the quiet room and the lactation room might need to be the same place even if this is not ideal). Following this acknowledgment, I know it is difficult to prioritize some needs over others, but I wonder if it is possible to classify some of these recommendations as essential or highly desirable, or some other form of categorization providing guidelines when there are constraints limiting our ability to have it all.

An additional recommendation I would make: when deciding on social events to attend with a big crowd from the conference, consider the identities of people in the group and respect local community spaces for queer and trans people, especially queer and trans people of color (more than once I have been in a local queer bar during a conference, just to have a large number of cis, straight and white scientist crash the place and immediately change the mood).

I have two additional bigger-picture thoughts that I believe should not play into the decision to accept this manuscript: 1. This piece is much broader than microbiology conferences and this might be an odd venue for it. I understand that there are not that many venues for pieces like this, and that this issue is a good fit, however, I worry that pitching it in the microbiology space could restrict the readership.

2. My second thought is about the need to write this as a peer-reviewed piece. I understand that in academic spaces peer-review publications are the currency and we are fairly bad at giving good credit for other contributions. This document, to me, however, reads more like a white paper that should be readily available and in a more flexible format so that it can be periodically updated in consultation with different queer scientists and activists. I wonder if peer review is too small of a forum for these kinds of discussions. Maybe we need more open spaces for the community to engage (in a respectful manner).

Response to reviewers:

Reviewer #1 (Comments for the Author):

Review of manuscript:

Building a Queer and Trans-Inclusive Microbiology Conference

By Rachel Gregor et al.

Review by Jennifer Middleton

This work presents a thoughtful, well-rationed, and easy to follow set of suggestions for conference organizers to consider when aiming to host conferences that are safe and inclusive of queer and trans attendees. The suggested guidelines represent a synthesis of the authors' diverse conference experiences and build upon the growing body of literature investigating the LGBTQ+ experience more broadly. Importantly, this work gives careful consideration to intersectionality within the LGBTQ+ community and several of the recommended actions (including consideration of local climate, confidential reporting, travel awards for low-income attendees, and on-site quiet spaces) will benefit a wide-range of conference attendees outside of the LGBTQ+ community as well. Although this work is targeted for the microbiology community, many of the recommendations are easily transferrable to other disciplines. Further, this work includes in its appendix an additional compilation of helpful resources and guides that will be quite useful to anyone interested in building a more inclusive STEM environment. As such, I believe this work provides a valuable framework for queer and trans-inclusion both in the microbiology community and across STEM disciplines.

General Comments:

I appreciate the authors' efforts to motivate and produce a clear and actionable set of recommendations for queer and trans-inclusion at STEM conferences. I especially appreciate the authors' call to consider regional and cultural nuances in alleged queer-friendly spaces, patience in the adoption and use of evolving language and terminology, and attention to avoiding white-bias when considering LGBTQ+ experience. This is already a valuable and important contribution. Below I have listed a few specific suggestions to help clarify a few of the main points from the text for a wider audience.

We thank the reviewer for their support and encouragement of this work. Please find our responses to specific comments below.

Specific Comments:

Abstract: I appreciate that this article is written primarily by and targeted for the microbiology community. However, given the number of suggestions that are clearly applicable across STEM fields (and beyond), I recommend the authors highlight within the abstract the utility of this work for applications outside of microbiology as well.

We thank the reviewer for their detailed comments and thoughtful review. We agree these recommendations can be useful outside of the microbiology community, but we think it is important that we are writing this piece as early career microbiologists in a microbiology journal published by a major microbiology society. We hope that this context means the piece will be seen as a call to action especially for more senior members of the field, and not be seen as something too general or outside of their area of expertise, which could lead to a loss of the central audience of our manuscript. However, we have re-written the abstract to emphasize that while the work is written by microbiologists, it is broadly useful in the sciences:

Lines 49-52: “Here, we draw from our experiences as early-career microbiologists to provide concrete, practical advice to help conference organizers across wider research communities design inclusive, safe, and welcoming conferences, where queer and trans scientists can flourish.”

Space limits restrict a more in-depth discussion in the abstract, but please see our related response to reviewer #2 below.

Lines 108-109: Perhaps this point is clear to those within the microbiology community, but from an outside perspective I think it would be helpful for the authors to clarify/elaborate on how personal financial precarity affects one's ability to attend professional conferences (which in my field are typically covered by research grants or other university funds). Alternatively, if this line is meant to address professional financial security (e.g., perhaps less funding allocated to queer and trans people), then I encourage the authors to clarify this point.

Thank you for this clarification. We intended to convey the first point brought up above, and we have added the following clarification:

Lines 107-108: “High-cost venues disproportionately discourage queer and trans attendees, who are more likely to face financial precarity (15, 16) meaning they may not be able to afford the upfront payments for later reimbursement as is common in many academic institutions.”

Lines 119-120: If the argument here is that travel awards are needed to help queer and trans attendees to attend conferences because they are financially disadvantaged, then it is not clear to me from this article why this suggestion is for specific awards for queer and trans attendees and not a blanket suggestion for more awards for attendees facing financial difficulties. If the argument is that specific awards for queer and trans people are necessary to increase queer and trans attendance (in the case that blanket financial awards typically overlook the queer and trans community, for example), then I believe that point needs to be better clarified.

We have added the following clarification:

Lines 120-122: “Awards based solely on financial need are likely to help queer and trans attendees as well, but may not be sufficient to counteract the various barriers these attendees face internationally and in academia that may not always be immediately visible.”

Space limitations on this piece prevent further discussion of these points, though we agree there is an interesting and important discussion to be had about how funding should be most effectively and equitably dispersed.

Lines 124-126. This is an important point and I appreciate the authors noting the nuance here.

Thank you for your encouragement on our presentation of this argument.

Lines 216-220: It might be worth making the point here that it can be difficult (or may feel unsafe) for queer and trans scientists to organically meet and identify each other in the professional environment outside the context of a dedicated event because this aspect of identify is not necessarily visually apparent and may not come up naturally in conversation.

Thank you for raising this point, we absolutely agree. We have added the following sentence noting it:

Lines 219-221: “Wide advertisement of LGBTQ+ events is critical, as it may not always be apparent which conference attendees are queer and/or trans, and attendees may struggle to connect with their community otherwise.”

Reviewer #2 (Comments for the Author):

This is a great compendium of best practices to increase inclusivity for queer and trans people in scientific conferences. At a time of intense backlash against queer and trans communities, it is a pressing piece. The paper is clear, holistic, and takes intersectionality seriously. As such, it is an important contribution that can be very helpful for committees organizing conferences.

My only comment in terms of improving the manuscript is to potentially acknowledge that these are best practices and that in some cases some might be in conflict. Having organized conferences outside of Europe and North America, I can say that sometimes the spaces available might not support a hybrid set-up and economic constraints might restrict the available extra spaces (that is, the quiet room and the lactation room might need to be the same place even if this is not ideal). Following this acknowledgment, I know it is difficult to prioritize some needs over others, but I wonder if it is possible to classify some of these recommendations as

essential or highly desirable, or some other form of categorization providing guidelines when there are constraints limiting our ability to have it all.

We thank the reviewer for their thoughtful and thorough comments on this piece. We absolutely understand that sometimes it will not be possible to achieve all these best practices and there will be a need for compromise under resource constraints. While we believe that the list of recommendations in this piece is achievable for larger conferences (200+ participants) in our field, it may not be possible to implement all these recommendations immediately, especially for smaller conferences that are not as well-resourced. We hope that these recommendations can serve as a starting point for conference organizers, to familiarize them with best practices that can then be implemented on a case-by-case basis. We hesitate to apply a ranking system to our recommendations, due to concerns that this will give the impression that some lower-ranked recommendations can be ignored, or that the needs of certain groups are more important than those of others, as the reviewer also points out.

We have added the following to the text that emphasizes that these recommendations are a baseline, and that conferences are encouraged to go above and beyond them:

Lines 277-283: “We recognize that it might not always be possible to implement all of these recommendations, or that some may occasionally be at odds with each other, requiring organizers to implement creative solutions in collaboration with queer and trans stakeholders from their research community. Nevertheless, we believe that the scope of these best practices is within reason for many conferences, and we strongly encourage organizers to prioritize these issues and continue in dialogue with queer and trans community members and needs and norms evolve in the future.”

An additional recommendation I would make: when deciding on social events to attend with a big crowd from the conference, consider the identities of people in the group and respect local community spaces for queer and trans people, especially queer and trans people of color (more than once I have been in a local queer bar during a conference, just to have a large number of cis, straight and white scientist crash the place and immediately change the mood).

We thank the reviewer for raising this issue. Having been in similar situations, both inside and outside of conference spaces (e.g., bachelorette parties at a drag show), we understand how frustrating it can be to have a space be occupied by a large group that does not understand or respect it. It is difficult to navigate the tension between making spaces inclusive while still maintaining their integrity. We worry about this in particular for closeted or recently-out members of the community, who we don't want to preemptively discourage from joining an event. Nevertheless, we do agree that it is important to make clear which events and spaces are for queer and trans attendees, and which events allies would be encouraged to attend. Space limits the depth with which we can discuss these points in the main text, but we have added the following:

Line 217-223: “Networking can be facilitated by planning queer and trans social and professional events, and including specific groups such as queer and trans People of Color (35, 36). Wide advertisement of LGBTQ+ is critical, as it may not always be apparent which conference attendees are queer and/or trans and attendees may struggle to connect with their community otherwise. It is important to note explicitly which events are for specific groups in particular and which events allies would be encouraged to attend.”

Line 225-226: “Consider off-site events at inclusive spaces for attendees’ safety and privacy, in consultation with local queer and trans organizers to ensure that any local community spaces are respected.”

I have two additional bigger-picture thoughts that I believe should not play into the decision to accept this manuscript: 1. This piece is much broader than microbiology conferences and this might be an odd venue for it. I understand that there are not that many venues for pieces like this, and that this issue is a good fit, however, I worry that pitching it in the microbiology space could restrict the readership.

While we understand that our choice of focus and venue may be slightly counterintuitive, it was important to us to write a call for action specifically within our own field. We are a group of early career microbiologists, speaking from our own experiences at microbiology conferences, in a microbiology journal published by one of the largest microbiology societies. While the content might be quite general, our perspective is not, and we intend this to be a call to action for those in our field. We believe that this specific context and venue will make the piece more relevant and engaging to those in our field than a very general piece would, even if it was in a bigger or more general journal. However, to make clear the larger scope of the piece, we have altered the abstract as follows:

Lines 49-52: “Here, we draw from our experiences as early-career microbiologists to provide concrete, practical advice to help conference organizers across research communities to design inclusive, safe, and welcoming conferences, where queer and trans scientists can flourish.”

2. My second thought is about the need to write this as a peer-reviewed piece. I understand that in academic spaces peer-review publications are the currency and we are fairly bad at giving good credit for other contributions. This document, to me, however, reads more like a white paper that should be readily available and in a more flexible format so that it can be periodically updated in consultation with different queer scientists and activists. I wonder if peer review is too small of a forum for these kinds of discussions. Maybe we need more open spaces for the community to engage (in a respectful manner).

We appreciate this perspective and indeed considered these issues when finding a format and venue for publishing this piece. It was extremely important to us that this piece be readily available, which led us to submit to *mSystems* which is an open access journal, and to preprint this work as well on *EcoEvoRxiv* simultaneously to submission. We also considered how to widely collect community feedback beyond the standard peer review process. To this end, in the process of conceptualizing this piece we attempted to reach as many queer and trans microbiologists as we could find to invite them to participate. This work incorporates perspectives from the 15 main text authors, 14 Queer and Trans in Microbiology Consortium level authors, 5 additional individuals who provided feedback to the manuscript, and numerous other community members who provided informal feedback during conferences in the preceding year.

For the reasons that the reviewer notes above, and in order to have a concrete document to refer back to and reach the community, we did conclude that a peer reviewed journal article would be the best available venue for this piece. We agree that the best practices described here will certainly change in the future, and it is crucial to create spaces to continue these conversations. We are extremely excited to see the new Pride in Microbiology Network which has been recently announced (<https://doi.org/10.1038/s41564-023-01394-y>) and hope that this will provide such a space within our field. We have added a citation (#37) and updated footnote 8 (pg 14) to highlight this initiative:

“In 2021-2023, LGBTQ+ events have been held at: Marine Microbes GRC, ISME, American Society for Microbiology (ASM) Microbe, the Australian Society for Microbiology (ASM) Annual National Conference, the Australian Microbial Ecology (AusME) annual meeting, the Society for Integrative and Comparative Biology, the Microbiology Society Annual Conference, and the World Microbe Forum. Additionally, the Pride in Microbiology Network (<https://prideinmicrobiology.github.io/>) has been recently founded to provide a platform for these events year-round.”

We also note in the conclusion that norms and community needs may change in the future:

Lines 277-283: “Nevertheless, we believe that the scope of these best practices is within reason for many conferences, and we strongly encourage organizers to prioritize these issues and continue in dialogue with queer and trans community members as needs and norms evolve in the future.”

August 8, 2023

Dr. Rachel Gregor
Massachusetts Institute of Technology
Civil and Environmental Engineering
Cambridge, MA 02141

Re: mSystems00433-23R1 (Building a Queer- and Trans-Inclusive Microbiology Conference)

Dear Dr. Rachel Gregor:

Your manuscript has been accepted, and I am forwarding it to the ASM Journals Department for publication. For your reference, ASM Journals' address is given below. Before it can be scheduled for publication, your manuscript will be checked by the mSystems production staff to make sure that all elements meet the technical requirements for publication. They will contact you if anything needs to be revised before copyediting and production can begin. Otherwise, you will be notified when your proofs are ready to be viewed.

If you would like to submit a potential Featured Image, please email a file and a short legend to msystems@asmusa.org. Please note that we can only consider images that (i) the authors created or own and (ii) have not been previously published. By submitting, you agree that the image can be used under the same terms as the published article. File requirements: square dimensions (4" x 4"), 300 dpi resolution, RGB colorspace, TIF file format.

We recognize that the video files can become quite large, and so to avoid quality loss ASM suggests sending the video file via <https://www.wetransfer.com/>. When you have a final version of the video and the still ready to share, please send it to mSystems staff at msystems@asmusa.org.

Sincerely,

Suzanne Ishaq
Editor, mSystems

Journals Department
E-mail: mSystems@asmusa.org